# Recurrent Versus Metastatic Head and Neck Cancer: An Evolving Landscape and the Role of Immunotherapy

**DOI:** 10.3390/biomedicines12092080

**Published:** 2024-09-12

**Authors:** Maria Paola Belfiore, Valerio Nardone, Ida D’Onofrio, Mario Pirozzi, Fabio Sandomenico, Stefano Farese, Marco De Chiara, Ciro Balbo, Salvatore Cappabianca, Morena Fasano

**Affiliations:** 1Diagnostic of Imaging, Department of Precision Medicine, Campania University ”L.Vanvitelli”, 80131 Naples, Italy; valerio.nardone@unicampania.it (V.N.); ida.donofrio@unicampania.it (I.D.); marc.dechiara93@gmail.com (M.D.C.); salvatore.cappabianca@unicampania.it (S.C.); 2SCDU Oncologia, “Maggiore della Carità” University Hospital, 28100 Novara, Italy; m.mariopirozzi@gmail.com; 3Radiology Unit, Buon Consiglio Fatebenefratelli Hospital, 80123 Naples, Italy; sandomenico.fabio@fbfna.it; 4Medical Oncology, Department of Precision Medicine, Campania University “L.Vanvitelli”, 80131 Naples, Italy; stefano.farese93@gmail.com (S.F.); cirobalbo@gmail.com (C.B.); morena.fasano@unicampania.it (M.F.)

**Keywords:** head and neck, locoregional disease, salvage surgery, reirradiation, locoregional treatment

## Abstract

Squamous cell carcinoma of the head and neck (SCCHN) is among the ten most common cancers worldwide, with advanced SCCHN presenting with a 5-year survival of 34% in the case of nodal involvement and 8% in the case of metastatic disease. Disease-free survival at 2 years is 67% for stage II and 33% for stage III tumors, whereas 12–30% of patients undergo distant failures after curative treatment. Previous treatments often hinder the success of salvage surgery and/or reirradiation, while the standard of care for the majority of metastatic SCCHN remains palliative chemo- and immuno-therapy, with few patients eligible for locoregional treatments. The aim of this paper is to review the characteristics of recurrent SCCHN, based on different recurrence sites, and metastatic disease; we will also explore the possibilities not only of salvage surgery and reirradiation but also systemic therapy choices and locoregional treatment for metastatic SCCHN.

## 1. Introduction

Squamous cell carcinoma of the head and neck (SCCHN) is among the ten most common cancers worldwide, with a mortality rate estimated at 350,000 per year, representing in Italy around 3% of all malignant diseases [1]. Fifty-four percent of cases present as advanced SCCHN at diagnosis, with a 5-year survival of around 34% in the case of regional node disease and 8% for metastatic disease.

While less than 5% SCCHN are already metastatic at diagnosis, 12–30% develop distant failures after curative treatment. Indeed, a large percentage of SCCHN patients develop treatment failure at 5 years: 5-year locoregional control is around 50%, whereas distant control, known as “freedom from distant metastasis” is about 85% [2]. In 5–7% of cases, recurrence occurs in neck lymph nodes. A fundamental distinction must be made between locoregional recurrence and second primary tumor, described in 14% of cases. Locoregional recurrence is the main limiting factor for long-term survival, being, in some cases, the main cause of death. In fact, treatment is complicated both by the effects of previous therapies received and because recurrence is often typically multifocal and infiltrative.

Recurrent disease still represents a challenging obstacle in SCCHN treatment. Salvage surgery or reirradiation may be helpful in locoregional recurrence, but previous curative treatments may hinder their success or increase the possibility of adverse events. Systemic therapy can be used both in recurrent and metastatic settings, with a median overall survival (mOS) of 13.6 months for the chemo–immunotherapy combination in first-line CPS ≥ 1 SCCHN; on the other hand, locoregional approaches can be used in oligometastatic patients to delay the start of systemic treatment and obtain a significant increase in survival.

Imaging in SCCHN is crucial for detecting or excluding tumors, delineating their extent, including the involvement of adjacent structures, bone, nerves, lymph nodes, and identifying distant metastases. It guides biopsies, radiotherapy planning, and computer-assisted surgery, evaluates therapy outcomes, and detects tumor persistence or recurrence for salvage therapy initiation. Modern cross-sectional imaging modalities significantly contribute to pre- and post-therapeutic evaluation, treatment planning, and establishing prognosis [3].

Our work aims to review the different characteristics of locoregional recurrence and metastatic disease, characterized by a different symptomatic burden, clinical features, prognostic impact, and diagnostic and therapeutic strategies.

## 2. Locoregional Disease

Regarding locoregional disease, one must first differentiate between persistent disease, second primary tumor and real locoregional recurrence [4,5]. Persistent disease can be defined as the absence of disease control within 6 months of curative treatment. To consider a malignancy as a second primary tumor, malignant tissue must be 2 cm apart from and/or must develop at least 3 years after the first tumor. In fact, we differentiate between synchronous (when a second primary tumor develops within 6 months after the first diagnosis) and metachronous tumors (when more than 6 months have passed since the first diagnosis). Lastly, local recurrences develop within 2 cm from and within 3 years after the first tumor. Difference in timing and localization is important, and it is based on the “field cancerization” concept, which states that the adjacent normal tissue often harbors preneoplastic characteristics that more easily lead to local recurrence or second primary tumors [6].

Recently, several studies have analyzed the genetic profile or the multi-omics pattern of SCCHN samples, e.g., in 2015, De Cecco et al. proposed a classification of all head and neck cancers into six clusters based on activated functional pathways and molecular and biological characteristics [7], while different genomic analyses identified alterations that may serve as markers of increased risk of recurrence, with multi-omics analysis identifying possible biomarkers for risk stratification [8,9,10]. Indeed, more widespread and affordable technologies may facilitate differential diagnosis between second primary and local recurrence.

CT is typically the first modality ordered for assessment of SCCHN locoregional recurrences due to its availability, cost-effectiveness, and shorter scanning times compared to MRI. CT offers high-quality multiplanar reconstructions and superior evaluation of bony structures. However, it uses ionizing radiation and provides poorer soft tissue contrast than MRI. MRI is the best option for head and neck imaging and, to define its extent, offers higher soft tissue contrast, evaluates blood vessels without contrast media, and avoids radiation exposure. Functional MRI sequences like DWI and perfusion scans provide specific lesion characterization, though MRI is more costly and less tolerated by some patients due to longer examination times and potential artifacts [3].

In our paper, we decided to focus on second primary tumors and local recurrences, both with their distinctive characteristics and treatment options. Of course, salvage treatment and definite treatment for a second primary strictly depend on what treatment was chosen for the first primary. For early-stage disease, surgery or radiotherapy are both equally valuable options with similar LCR [11]. Radiotherapy allows to bypass surgical risk, but it may carry long-term risks, such as swallowing dysfunction, xerostomia or hearing loss. Organ-preserving surgery could lead to bleeding, nerve injury and dysphagia but present reduced long-term toxicity when compared to RT, which is also confirmed by the latest ORATOR result [12,13]. In the case of a locally advanced disease, single-modality treatment is not enough, and the choice rests between surgery, followed by adjuvant radiotherapy with or without chemotherapy, or concomitant chemoradiotherapy (CRT). Surgery is often indicated for an advanced oral cavity and laryngeal cavity, while CRT is often preferred for oropharyngeal cancer. Of course, CRT is the treatment of choice when a disease is unresectable or when the price of radical surgery seems abysmal compared to the outcome [11,14].

### 2.1. Second Primary

Not all head and neck tumors have the same frequency as second primary tumors. León et al. demonstrated in 2012 that incidence is different between first and successive tumors according to their location [15]: laryngeal cancer incidence tends to decrease with each primary tumor (51.6% for first primary, 25.7% for second primary and 13.3% for third primary tumors), whereas oral cavity and oropharyngeal neoplasms presented with increased incidence in each successive tumor (e.g., 11.6% for first to 35% for third primary oral cavity tumor and 17.5% for first to 31.7% for third primary oropharyngeal cancer). Furthermore, the earlier stages percentage appears to increase with each successive tumor: 56% of first SCCHN are T1-T2 tumors, but this reached up to 63.4% for second and 71.7% for third primary tumors.

León et al. also demonstrated that an increasing number of patients with each new primary are not suitable for radical treatment; in fact, while around 64% of first primary tumors were treated with radiotherapy, with or without chemotherapy, only 24% of second primary tumors were eligible.

Survival rates appear to be influenced by previous treatment: Jones et al. [16] and Farhadieh et al. [17] both found no difference in survival after previous radiotherapy, whereas Robinson et al. demonstrated better survival rates for patients who had not received radiotherapy. In the study of Dolan et al. [18], 61% of patients who developed a second neoplasm in the irradiated field did not gain a significant advantage from locoregional treatment, versus only 30% of those who did not receive radiotherapy.

Radiomic, clinical, and volumetric models each provide valuable insights while managing second primary tumors. Significant correlations between apparent diffusion coefficient (ADC) parameters on magnetic resonance imaging (MRI) scans and local outcomes have been demonstrated when diffusion-weighted (DW) imaging is performed on SCCHN in the treatment course. Two weeks post-treatment initiation, tumors with higher histogram skewness, higher histogram kurtosis, and a smaller increase in the percentage change in mean ADC are significantly more likely to exhibit local failure than local control. DW imaging-based prediction of tumor resistance to chemoradiotherapy (CRT) could pinpoint patients needing more aggressive post-treatment investigations to detect residual cancer, allowing earlier salvage surgery. The goal is to use DW imaging prospectively to identify patients who might benefit from treatment modifications, such as a radiation therapy boost, additional targeted therapies, or a switch from CRT to surgery. Tumors with a smaller increase in the percentage change in mean ADC two weeks after treatment initiation are more likely to show local failure. Future models that combine mean ADC, skewness, and kurtosis data may improve the accuracy of DW imaging. Deep learning models can effectively predict death prognosis and cancer recurrence using GTV (Gross Tumor Volume) and PTV (Planning Target Volume) radiomics data [19].

### 2.2. Locoregional Recurrence

On the other hand, recurrent disease tends to be increasingly infiltrative and multifocal, often presenting with microscopic deposits that appear outside the field subjected to radiotherapy or surgery and therefore not easily found with common imaging methods. Moreover, it features a higher degree of perineural invasion [20]. The diagnosis is often not only difficult and late but also often complicated by post-radiation or post-surgery fibrous tissue or even the altered anatomy after the primary treatment. All these characteristics can be the cause of the failure of the proposed treatment.

The prognosis of locoregional recurrence is affected by prognostic factors linked to the patient (poor performance status, comorbidities, age, weight loss > 5%) and linked to the tumor (advanced recurrent stage III and IV, short disease-free survival DSF, previous chemotherapy).

Locoregional recurrence control is critically linked to patient survival. A study by Shakir et al. [21] revealed that neural network analysis of radiomics features in head and neck cancer yielded promising results in tumor histology classification. Predictive models trained to forecast death prognosis and cancer recurrence using GTV radiomics features achieved notable classification accuracy. These models showed improved sensitivity and specificity compared to previous research using clinical factors such as nodal stage, tumor size, stage, resectability, and hemoglobin levels to predict 2-year survival. Advances in technology are expected to further enhance the accuracy of predictive models, thereby improving support for treatment decision-making [22].

A study by Chang et al. [23] exclusively on locoregional recurrent SCCHN demonstrated different recurrence rates for different sites. Recurrence site is one of the main prognostic factors, influencing disease presentation, symptoms, treatment choice and survival. The recurrence rate (RR) by Chang et al. is 15.45% for oral cavity, 11.05% for oropharynx, and 9.9% for hypopharynx. Of course, increased stage of recurrent tumor presents with reduced survival: 2-year DFS for stage II recurrent tumor is around 67% versus 33% for stage III recurrent cancer [24].

However, Zbären et al. [25] demonstrated that recurrent laryngeal cancer is often under-staged due to residual inflammation or functional post-radiotherapy changes. In their work, 52% of recurrent glottic carcinomas were under-staged, and only three cases were considered over-staged. It is possible that this could be the case for all head and neck tumors.

Of course, extensive recurrences [26] and shorter disease-free intervals [27,28] are associated with worse outcomes.

#### 2.2.1. Laryngeal Recurrence

Laryngeal recurrences are characterized by earlier presentation and higher symptomatic burden; however, they are often more localized than other types with a reduced probability of positive lymph nodes or metastasis development due to limiting anatomical barriers and lack of lymphatic drainage.

Accordingly, salvage surgery is still a viable option, with high overall survival rates.

Total and partial laryngectomy are both valid alternatives. Although Rodriguez-Cuevas et al. [29] reported similar survival rates between total and partial salvage laryngectomy, others demonstrated poorer survival and reduced disease control in patients who needed salvage total laryngectomy: Ganly et al. [30] presented an mOS rate of 88 vs. 55%, respectively, for partial and total laryngectomy, while Holsinger et al. [31] also found increased survival for conservative laryngectomy with no difference in local recurrence rate. Furthermore, partial laryngectomy is well tolerated despite previous radiation therapy: overall complication rate [30,32,33] is around 19–28%. Investigators agree that, after establishing proper selection criteria for partial laryngectomy, functional surgery may be an appropriate therapeutic choice. However, it must be noted that partial laryngectomy is available in the case of early-stage recurrences, thus limiting comparison between the two techniques.

#### 2.2.2. Oral Cavity Recurrence

Oral cavity recurrences are easier to detect; indeed, self-reported symptoms such as pain, burning sensation, difficulty in chewing and swallowing, and globus, have been found to help detect early recurrence in up to 78% of cases [34]. Clinical examination can also easily identify signs of recurrence, such as ulcers, lumps, eroded areas, etc. Higher grade and lymph node status are significantly related to tumor recurrences, whereas there are contrasting results for tumor size [34,35,36,37,38]; tongue localization [39] and comorbidities [40] also seem to be important prognostic factors.

However, survival in oral cavity cancer is among the poorest: 5-year OS for oral cavity recurrence is around 30% [39]. Disease-free interval (DFI) appears to be a significant prognostic factor for recurrence: several studies demonstrated that a longer DFI is associated with more indolent and treatable recurrences [41,42,43]. However, the recurrence interval is not a fixed entity, and each author tends to define it differently, ranging from 6 weeks [44,45] to 3 months [46] or more [43,47].

Salvage surgery remains the standard treatment for locoregional recurrences, especially after previous radiotherapy. Patients treated with salvage surgery tend to have better prognosis and lower complication rates than those treated with reirradiation; this may also be due to the fact that early-stage recurrences are more frequently treated with surgery than advanced-stage recurrences [43,46,48].

However, reirradiation, with or without concomitant chemotherapy, may be a valid alternative, and indeed, several studies report 5-year OS rates of around 15–38% [27,49,50] despite being burdened by higher toxicity, such as soft tissue necrosis, nerve damage, trismus, risk of carotid blow-out, dysphagia, etc. [51,52] (Figure 1).

#### 2.2.3. Oropharyngeal Recurrence

In the RTOG 0192 trial, the locoregional recurrence rate for oropharyngeal cancer reached 14%. Oropharyngeal recurrences are unfortunately characterized by worse outcomes; treatment is not only difficult by itself due to organ location, e.g., close to other organs essential for breathing and deglutition, but it is often complicated by sequelae of previous treatments, e.g., altered anatomy, soft tissue fibrosis, restriction in jaw opening [53]. Indeed, salvage surgery generally presents with a 5-year OS rate of only 13–28% [51,52], and it is often associated with its own share of complications, such as dysphagia, aspiration and dysarthria. Indeed, Kim et al. [54] presented, in their trial with free flap transfer, a 34% rate of major complications; although no complete free flap necrosis occurred, we find facial or neck skin necrosis, salivary fistula, pneumonia, wound dehiscence and partial skin graft loss among the main complications.

Reirradiation has also been proposed as an alternative, with a 5-year OS reported by Zafereo et al. [52] of 32%; however, there is a high toxicity profile, with 9–32% of patients experiencing severe or fatal complications [51].

Further recurrences after salvage treatment remain a problem. Kim et al. [54] reported that 74% of patients developed an ulterior recurrence around 9 months later, while in the study by Zafaereo et al., 66% of patients developed a new recurrence after 8 months [52].

#### 2.2.4. Hypopharyngeal Recurrence

Hypopharyngeal recurrence survival and functional rates are worse than those for other head and neck sites and can remain asymptomatic for a long time. In fact, due to hypopharyngeal anatomy, it may grow along pharyngeal walls, piriform sinuses and the post-cricoid region without causing many symptoms. These often develop in the occasion of laryngeal invasion or nodal metastasis due to its lymphatic network draining to II–IV level cervical nodes and retropharyngeal nodes. It can also infiltrate or impinge close structures such as the esophagus and oropharynx, rendering the lesion unresectable. Hypopharyngeal tumors can thus present with dysphagia, sore throat, hoarseness, and globus sensation.

As more and more patients are treated on first occurrence with radiotherapy, with or without chemotherapy, surgery is often relegated to salvage treatment. However, Taki et al. [55] showed that, although most rT3 and rT4 patients were unsuitable for pharyngolaryngectomy, even in early recurrence (rT1, rT2), disease control was only obtained in around 26% of patients: a total of 70.7% of hypopharyngeal recurrence was not suitable for salvage surgery. As per post-operative complications, pharyngocutaneous fistula is one of the most troubling. Wakisaka et al. [56] found an increased incidence of pharyngocutaneous fistula in salvage surgery after chemoradiotherapy than in primary surgery; furthermore, the incidence was more than four times higher when the surgery occurred within 1 year from radiotherapy with or without concomitant chemotherapy. Duration of fistula and wound healing was also greater in patients who underwent salvage surgery in lieu of primary surgery and even greater in those who first underwent chemoradiotherapy. This is also confirmed by a meta-analysis by Paydarfar and Birkemyer [57]. Johansen et al. [58] also reported an increased fistula rate according to radiotherapy dose. However, Markou et al. [59] did not find any difference in fistula formation between those who underwent surgery as primary treatment or salvage treatment, irrespective of radiotherapy too (11.6 after radiotherapy vs. 13.3% without).

#### 2.2.5. Nasopharyngeal Recurrence

Nasopharyngeal cancer treated with intensity-modulated radiotherapy (IMRT) fortunately has an excellent control rate. A study by Au et al. [60] demonstrated that, out of more than 3000 patients, only 14% developed local recurrence or persistent disease, with a median interval of 30 months between first diagnosis and locoregional relapse. Locoregional recurrence in nasopharyngeal cancer tends to develop in the high-dose field mostly due to the radioresistance of the primary tumor [61,62]. Considering this, salvage surgery should be considered when treating relapsing nasopharyngeal tumors. While previous experience found similar results between surgery and reirradiation, more recently, in 2015, You et al. demonstrated increased OS (5-year OS 77.1 vs. 55.5%), improved quality of life and reduced complication rate with endoscopic nasopharyngectomy [63]. Regarding the issue of reirradiation, experts from major medical centers worldwide provided recommendations on re-irradiation [64], a task complicated by the proximity of the previous radiation dose to the tolerance limits of surrounding structures. These guidelines address various aspects of reirradiation, focusing on treatment strategy, target delineation, dose prescription, and dose constraint criteria for organs at risk (OAR). Systemic treatment remains a possibility, both concurrently or without radiotherapy: a case-control study by Liu et al. in rT3-4 relapsing nasopharyngeal cancer, treated with concurrent chemoradiotherapy or chemotherapy alone, reported no significant difference between the two groups’ 5-year OS rates.

#### 2.2.6. Nodal Recurrence

Besides strictly local recurrences, the majority of locoregional recurrences and treatment failures occur in regional nodes [65]. Several works have demonstrated how cervical nodes are the first site of treatment failures, especially after post-operative radiotherapy. Indeed, between 73% and 100% nodal relapses occur in the high-dose region [65,66,67]. Neck recurrences may also signal the possibility of distant metastasis [68]. Lim et al. [69] identified several prognostic factors for salvage treatment of neck recurrences; for example, a DFI of more than 1 year, stand-alone previous surgery, and recurrence in undissected neck or N1 have been found to be predictors of successful salvage surgery.

Positron emission tomography-computed tomography (PET/CT) shows focal increases in fluorodeoxyglucose F-18 (18F-FDG) glucose uptake, and bone destruction and erosions are well visualized on CT scans. Lymph node metastasis is an important prognostic factor, with guidelines suggesting specific size criteria for pathological nodes. Small metastatic nodes can show necrosis or increased enhancement on CT and MRI. Extracapsular spread, detected via shape irregularities and surrounding tissue infiltration, indicates tumor aggressiveness and poor prognosis, requiring different therapeutic strategies.

Defining tumor and target volumes for radiotherapy requires pretreatment diagnostic imaging using CT and MRI to segment images into target volumes for high-dose radiation while avoiding normal tissues. Consensus guidelines are in place for this process. Radiomics and artificial intelligence (AI)-related research are expected to further personalize radiation treatment by incorporating detailed image data into predictive models.

Post-therapeutic imaging in SCCHN is challenging due to treatment effects like edema, swelling, fibrosis, and necrosis. PET/CT is sensitive for detecting locoregional disease, distant lesions, and metachronous primary tumors, though positive results often require biopsy confirmation due to a high false positive rate. Dual-energy CT (DECT) improves tumor visibility, boundary delineation, and evaluation of thyroid cartilage invasion [3] (Figure 2).

### 2.3. Chemoreirradiation as Salvage Therapy

For most resectable recurrent SCCHN, salvage surgery remains the treatment of choice, with a 36–60% OS rate [28,49,70]. However, the eligibility for salvage surgery in patients with recurrent SCCHN varies, with rates ranging from 20 to 50% depending on the series, characteristics of the recurrence (local, tracheostomal, nodal), and prior therapeutic interventions [71]. In cases of unresectable SCCHN, reirradiation represents the exclusive treatment with curative intent.

Historically, reirradiation for local recurrence or a second primary tumor was discouraged due to concerns of excessive toxicity and suboptimal survival outcomes. However, modern radiotherapy techniques, such as intensity-modulated, stereotactic, and proton therapies, have shown control rates at least comparable to pre-intensity-modulated radiotherapy (IMRT) techniques but with significantly reduced side effects and enhanced safety, as reported in literature. Specifically, the pooled 2-year local control (LC) and OS rates were 53% and 46%, respectively. Furthermore, the incidence of toxicities ≥ grade 3 was 26%, while G5 toxicities were at 3.1%. These findings have changed the treatment approach for local recurrence of head and neck cancer, with an increasing reliance on reirradiation [72].

In the absence of confirmed guidelines, the strategy for reirradiation in recurrent SCCHN necessitates a highly individualized approach. Given its heterogeneity, the primary objective of any intervention is to balance potential disease control with the impact of therapy-related toxicities on the patient’s quality of life, ensuring they do not surpass the morbidity expected from a disease’s natural progression. Achieving this balance requires a detailed examination of several factors. Among the patient-related parameters are comorbidities, the sequelae of initial treatments, and tube dependence. Regarding the tumor, factors such as its extent (as either a focal nidus or a more expansive volume-infiltrative spread), its histological subtype, and the type of recurrence (nodal or mucosal) should be addressed. Furthermore, a meticulous review of the previous radiotherapy is essential, emphasizing the relationship between the recurrence and previous radiation areas, whether in-field or out-of-field, and the organ at risk (OAR) constraints related to the previous treatment.

The optimal candidate exhibits a favorable performance status, younger age, early-stage tumor presentation, absence of bulky disease or organ dysfunction, etiological association with nasopharynx or HPV malignancies, a prolonged interval between radiotherapy courses, and an absence of significant radiation-induced adverse effects such as osteochondroradionecrosis, carotid artery stenosis, marked fibrosis or fistula formation, and myelopathies. In the absence of definitive evidence, nomograms have been introduced to aid in patient selection and the choice of an appropriate therapeutic strategy.

Riaz et al. constructed a model using prognostic indicators like recurrent stage, non-oral cavity subsite, lack of organ dysfunction, previous salvage surgery, and radiation doses exceeding 50 Gy, all significantly enhancing loco-regional control (LRC) [73]. Tanvetyanon and colleagues formulated a nomogram to predict mortality within two years after reirradiation, including indicators such as organ dysfunction, Charlson index, interval from previous radiation, recurrent tumor stage, bulky tumor, and the reirradiation dose [74].

In 2018, the American Multi-Institutional Re-Irradiation (MIRI) group published three pivotal articles to address the complexities of treatment selection in HNC recurrence. The first article [75] introduced a risk-stratification model utilizing recursive partitioning analysis (RPA). This study comprised 412 patients from seven institutions, all treated with reirradiation, either definitive or adjuvant, with a minimum dose of 60 Gy. They were categorized based on three determinants: the duration between radiation courses (over 2 years vs. within 2 years), the execution of salvage surgery, and the presence of organ dysfunction. Class 1, comprising patients with an extended interval between radiations and those undergoing salvage surgery, achieved a 2-year OS rate of 61.9%. Class 2, with a 40% OS rate, included those unfit for surgery but without organ dysfunction. The group with minimal time between treatments and manifest organ dysfunction, designated as Class 3, observed the lowest 2-year OS, standing at 16.8%. The findings underscored the association between the persistence of gross residual disease post-surgery and suboptimal survival outcomes. The subsequent article [76] further investigated specific aspects of reirradiation, exploring the details of treatment planning, including fractionation, total dose, and treatment volumes. The third article [77] notably validated the RPA classification for patients with unresected recurrences and compared the outcomes between those undergoing IMRT and SBRT treatments. To address the lack of evidence in this field, a recent study [78] reviewed 274 publications, focusing on key questions regarding patient selection, adjuvant and definitive reirradiation, stereotactic body radiation therapy (SBRT), and reirradiation for non-squamous cancers, providing updated evidence-based recommendations. The committee emphasizes the importance of careful patient selection for curative intent salvage therapy, advocating for multidisciplinary review and patient care goals. For resectable cancers, surgical treatment is strongly recommended, while systemic therapy alone is usually not considered sufficient. For unresectable diseases, fractionated reirradiation to 60–70 Gy along with concurrent systemic therapy is recommended. The role of SBRT remains unclear, with no consensus on its appropriateness compared to fractionated reirradiation. Even though concurrent chemoradiotherapy is often used as first-choice treatment for primary unresectable head and neck tumor, reirradiation can also be combined with chemotherapy in recurrent SCCHN to improve its benefit. A trial of cisplatin and paclitaxel concurrent to radiotherapy, RTOG 9911 [79], published in 2007, demonstrated significant results in OS and progression-free survival (PFS), with 2-year OS of 25.9% and a 2-year PFS of 15.8%. Results are improved when compared to those of a previous trial, RTOG 9610 [28], with 5-fluorouracil and hydroxyurea (2-year OS 15.2%). Significantly, while RTOG 9911 found no difference in survival according to time to relapse, patients in RTOG 9610 who underwent a previous course of radiotherapy within one year presented with worse survival. In RTOG 9911, around 85% of patients developed grade 3 to 5 toxicity. Main adverse events were myelosuppression (21% of patients, grade 4 or worse), infection and/or febrile neutropenia (15%, grade 3 or worse), gastrointestinal toxicity (48%, grade 3 or worse) and mucositis (14%, grade 3 or worse).

Unfortunately, the true merits of the addition of chemotherapy concomitant to reirradiation could only be demonstrated by a prospective phase III trial. RTOG 0421 was designed with this problem in mind, but it was closed earlier because of poor accrual. Clinical trials are currently exploring the use of immune checkpoint inhibitors (ICIs) in conjunction with re-irradiation for patients with locoregionally recurrent head and neck squamous cell carcinoma (SCCHN). The REPORT trial (NCT03317327) is assessing the safety and tolerability of nivolumab combined with hyperfractionated re-irradiation. The study will also evaluate efficacy outcomes, including PFS, OS and objective response rate (ORR). A parallel study (NCT02289209) is investigating a similar approach using pembrolizumab.

Indeed, even in newly diagnosed locally advanced SCCHN, immunotherapy is still limited to clinical trials, with dismal results. The PembroRad trial, comparing pembrolizumab + radiotherapy to cetuximab + radiotherapy, did not meet its primary endpoint of locoregional control at 15 months, with no difference in survival between the two arms [80]; on the other hand, the addition of pembrolizumab to cisplatin-based chemoradiotherapy in the Keynote 412 did not result in a statistically significant benefit in event-free survival [81]. However, while an earlier trial by Lee et al. reported no significant benefit from the addition of concomitant and sequential Avelumab to the standard of care [82], Zandberg et al. first presented improved OS and PFS and locoregional control with the use of sequential Pembrolizumab after chemoradiotherapy [83]. Thus, at the moment, immunotherapy is only limited to recurrent SCCHN not amenable to curative treatments.

Post-salvage surgery reirradiation and concomitant chemo-reirradiation is also a possibility. However, post-operative reirradiation is often limited to high-risk patients [78], e.g., in the case of extracapsular extension or positive margin. Janot et al. [71] evaluated radiotherapy concomitant to 5-fluorouracil and hydroxyurea chemotherapy after salvage surgery compared to surgery alone, demonstrating increased locoregional control and disease-free survival; unfortunately, no benefit in OS was found. Toxicity was also increased, with 40% of patients developing grade 3 or 4 late toxicity within 2 years from treatment (Table 1).

Patient selection is thus of primary importance; salvage surgery, previous chemo-radiotherapy, and an interval from previous radiotherapy equal to or over 36 months have been found to significantly influence overall survival [84]. The results are contrasting regarding patient-related factors. Choe et al. found no influence on overall survival, whereas in the study by Tanvetyanon et al. [74], organ disfunction and comorbidity were useful in predicting survival. It must also be noted that in Choe et al.’s work [84], the majority (80%) of patients presented with an ECOG performance status between 0 and 1, thus limiting the potential evaluation of comorbidities.

## 3. Metastatic Disease

Distant metastases have a 10% incidence at initial presentation, but 20–30% of patients will develop distant metastases later in their disease course. Once metastatization occurs, survival is scarce, with poor prognosis and a median overall survival of 10 months. Median time to metastasis is around 16 months [85,86]. The most common sites are lung and bone, later followed by liver, brain and skin. Remarkably, there is a higher frequency of bone metastasis in nasopharyngeal cancer than other head and neck tumors [87]. However, it is worth classifying metastatic patients into polymetastasic and oligometastatic. While for the first case, standard-of-care treatment involves systemic therapy with chemo- and immuno-therapy, oligometastatic disease, currently defined as five or less clinically detectable lesions [88,89,90], may be subjected to aggressive local treatment promoting increased survival and cure [89,91].

Several risk factors for the development of distant metastases have been identified, i.e., tumor site, stage, grade, residual disease, and ENE are the main known factors.

Oropharynx and hypopharynx are the most common primary sites for metastasis due to the important lymphatic network. Duprez et al. [85] found a 20.5% incidence for hypopharynx and 12.9% for oropharynx cancer, while Leon et al. [86], respectively, found a 16 and 7% incidence. Oral cavity tumors present a significantly lower risk, for example, in Leon et al. [86], the incidence amounts to 1%, with similar results found by Kuperman et al. [92] (1.95%) and Liu et al. [87] (1.8%); although, in Duprez et al. [85], it reached 15.4%.

Furthermore, HPV status plays a significant role, with HPV+ oropharyngeal cancers presenting with a reduced frequency of distant metastases [85]. The metastases from HPV+ tumors tend to manifest later and exhibit atypical behavior, often adopting a “disseminating” phenotype, which involves the spread to multiple organs, including multiple lung metastases. Unconventionally, these metastases may also target unusual sites such as the brain, skin, and abdominal nodes. Furthermore, patients with HPV+ tumors may experience prolonged survival after salvage treatments for distant metastases, indicating a more favorable response to interventions [93,94].

T and N staging is a significant risk factor: T3 and T4 present a risk of metastatization of 13–17% and 15–21% respectively, compared to 8–13% for T2 tumors [85,95,96]; on the other hand, N3 tumors present a risk of 19–29% compared to N2 (14–26%) and N1 (11–22%) [85,86,95].

Extranodal extension (ENE) also concurs to the development of distant metastasis; in fact, in Duprez et al. [85], a 20% increase was identified in metastasis rate between ENE- and ENE+ tumors (11.9 vs. 32%).

Other nodal characteristics have also been identified as predictive of distant metastasis: three or more metastatic nodes, bilateral metastatic nodes, nodes with a diameter of more than 6 cm or nodes in the lower jugular chains, and locoregional recurrence [97,98].

### 3.1. Oligometastatic Disease and Local Treatments

As anticipated, oligometastatic disease can be treated by local ablation treatments, mainly surgery and stereotactic ablative radiotherapy (SABR), although evidence mostly comes from retrospective studies [99,100,101,102,103]. SABR allows, non-invasively, for high rates of tumor control thanks to the precise delivery of a high radiation dose in a small number of fractions (1 to 10) [104].

A meta-analysis published in 2015 by Young et al. [101], including 11 studies and 387 patients, demonstrated a 29% 5-year OS in SCCHN patients who underwent surgical resection in metachronous pulmonary metastasis. The researchers also identified several negative prognostic factors, such as cervical lymph node metastasis at diagnosis, oral cavity primary site, incomplete resection, and multiple pulmonary nodules. A few years before, Haro et al. [105] published their 27 years’ experience at Kyushu University Hospital in SCCHN patients treated with pulmonary metastasectomy, presenting a 3- and 5-year OS, respectively, of 53.3% and 50%. A few other retrospective studies also demonstrated a significant benefit in SCCHN patients with oligometastatic disease (more than 50% of cases involved lung metastasis) treated with surgery or SABR, reaching a 40–50% 5-year OS [99,100,102,103,106].

Data comparing surgery and SABR are scarce, although SABR is frequently preferred due to its high safety [107], reduced invasiveness, flexibility and efficacy [108]. It can treat complex and moving targets, with fewer than 5% of patients experiencing grade 3 toxicity [109], and local control rates (LCR) of 70–90% at 2 years, depending on the series [100,103,110], regardless of dosage or technique. Particularly for HPV-negative patients with significant comorbidities, this method is well-tolerated and feasible, allowing systemic treatments to continue without interruption. Bonomo et al. found that SBRT for lung-only oligometastases in head and neck cancer patients could defer systemic treatment, with a median time to progression of 10 months and minimal acute toxicity (no reported G ≥ 3 toxicities) [100]. A retrospective analysis of pulmonary metastasis treated with SABR revealed a 2-year LCR of 94% and an OS rate of 62%, with no reported toxicities of grade equal to or greater than 3 [110]. In a study by Franzese et al. [103], SBRT for up to five oligometastases in head and neck cancer, primarily affecting the lungs (59%), achieved 1- and 2-year LCR of 83.1% and 70.2%, respectively, with 1- and 2-year OS rates of 81.0% and 67.1%. The treatment was well-tolerated with few toxicities, underscoring the importance of early intervention and careful patient selection. Factors such as prior local therapy, oligoprogression, and untreated metastases were identified as having a significant impact on local control, highlighting the importance of early intervention and careful patient selection.

While the previously mentioned experiences all came from retrospective studies, in 2019, Palma et al. [111] published the results of the randomized phase 2 SABR-COMET trial, in which 10% of the 99 total patients presented with metastatic SCCHN and an increased 5-year OS (42 vs. 17%) was demonstrated in patients who underwent SABR versus standard of care. A phase II trial by Sutera et al. [112] also found an increased survival rate in patients who underwent SABR, reaching a 43% 5-year OS in all patients and in the subgroup of SCCHN patients (10% out of 147 patients) and a 42% 5-year OS with a median survival of 18 months.

Ongoing trials are exploring the combination of SABR with systemic treatments [113]. Three phase 3 randomized trials (SABR-COMET-3, SABR-COMET-10, OligoRARE) are assessing the benefits for any oligometastatic primary sites. Additionally, numerous phase 1 and 2 single-arm trials are currently investigating the combination of radiotherapy with immunotherapy and chemotherapy. The GORTEC 2014-04 phase IIR study (OMET trial) evaluated the efficacy of SABR as upfront treatment for 1–3 pulmonary metastases in oligometastatic SCCHN patients, omitting frontline chemotherapy. The findings indicated a reduction in severe toxicity rates, with comparable survival rates and 1-year overall survival without quality-of-life deterioration.

While most locoregional approaches are related to lung metastasis, metastases in other sites can also be treated with local techniques.

Bone metastases in SCCHN are usually part of a systemic progression and not isolated metastases, appearing in conjunction with locoregional relapse and/or visceral dissemination [114]; however, bone-exclusive metastases are not uncommon, with a 24–46% incidence rate (24–50% of cases present with monostotic metastasis). Bone metastases incidence is now up to 1.9–8.4% thanks to the introduction of more accurate imaging such as FDG PET [115,116]. SBRT has been frequently used in solitary bone lesions to improve not only tumor control but also pain reduction in several tumor types. A study by Grisanti et al. has demonstrated how bone metastases treatment influences clinical outcome; indeed, bone radiotherapy has a statistically significant effect on bone metastases-OS (BM-OS), while bisphosphonates and denosumab are associated with increased survival only in the nasopharyngeal cancer cohort. Furthermore, radiotherapy alone in NPC patients with limited bone metastases also provided a survival benefit [114].

Few data can be found about liver metastasis, mostly focusing on squamous cell carcinoma of several origins or exclusively nasopharyngeal cancer. A study by Pawlik et al. published in 2007 reported a 5-year disease-free survival (DFS) of 18.6% and 5-year OS of 20.5% in liver metastasis by squamous cell carcinomas (only 12 out of 52 patients were SCCHN) treated with locoregional approaches [117]; in 2021, Kurihara et al. reported their results on 11 patients (5 patients had oropharyngeal cancer, 6 esophageal cancer), with a 45% 1-year recurrence-free survival, 72% 1-year OS and a 10-month DFS after hepatic resection; furthermore, patients who underwent resection after 1 year of systemic treatment presented with better results [118]. Partial hepatectomy also seems more promising than trans-hepatic arterial chemoembolization (TACE) in NPC patients, as OS in the former group was better in the study by Huang et al.; however, while reduced incidence of HBV infection in NPC patients allows for wider resection margins, patients who underwent resection reported more adverse events than the TACE group [119]. More recently, in 2022, Feng et al. reported a survival benefit in NPC treated with hepatic resection, with a mOS of 32.6 months against 19.57 months of the non-hepatic resection group [120].

Since 1953, reports of an “abscopal effect”—a radiotherapy-induced tumor reduction outside the irradiation field—have been published in several tumor types, although it is still regarded as a rarely occurring phenomenon [121]. In recent years, interest in the abscopal effect has resurfaced after the development of immune checkpoint inhibitors (ICI). Beyond direct cytotoxicity, it is speculated that the immune effects of irradiation may contribute to the so-called abscopal effect; however, radiotherapy influence on the immune system may not be as clear as initially thought. Irradiation can induce immunogenic cell death and can increase antigen presentation and consequent death by cytotoxic T cells; on the other hand, radiation-induced leukopenia may be a negative prognostic factor during treatment with ICIs (Table 2).

In SCCHN, the abscopal effect remains anecdotal, with only a few published case reports [122,123,124]. Bahig et al.’s trial in metastatic SCCHN patients treated with SBRT and the off-label durvalumab and tremelimumab resulted in a 6moPFS primary endpoint of 69.7% [125]. The phase II trial by McBride et al. of Nivolumab with or without SBRT demonstrated no improvement in ORR with no evidence of any abscopal effect [126].

### 3.2. Systemic Treatment

In patients not amenable to locoregional treatments, the standard of care remains systemic treatment. Decision on the treatment of choice is made depending on the patient and tumor characteristics. Tumor characteristics such as Combined Positive Score (CPS) status, platinum sensitivity, HPV status, tumor burden and disease pace play a pivotal role in treatment selection.

In platinum-sensitive patients, the first choice is usually a platinum-based combination with or without immunotherapy, whereas in platinum-resistant patients, immunotherapy or single-agent chemotherapy are the available alternatives [98].

In a combined positive score (CPS) < 1, platinum-sensitive SCCHN, the EXTREME regimen and its variations remain the treatment of choice. In CPS > 1 SCCHN, the physician can choose between single-agent pembrolizumab or a combination of chemotherapy and immunotherapy (Platinum + 5 fluorouracil + pembrolizumab). Chemoimmunotherapy is continued for 4–6 cycles followed up by pembrolizumab maintenance [11,127,128].

#### 3.2.1. The Role of Immunotherapy

Food and Drug Administration (FDA) and European Medical Agency (EMA) have approved both Pembrolizumab alone and Pembrolizumab associated with chemotherapy in all patients with CPS equal to or higher than 1 [129]. However, no direct comparison between Pembrolizumab and Pembrolizumab-CT has been made in Keynote-048. While no CPS cut-off exists for SCCHN as in other tumors (e.g., lung cancer), CPS levels remain an important junction in the treatment algorithm. No significant benefit in OS and RR has been demonstrated with Pembrolizumab against EXTREME in the subgroup CPS 1–19 (HR 0.86), while HRs favored Pembrolizumab and chemotherapy association in all subgroups (HR0.71 in CPS 1–19), independently of CPS status [127]. A negative CPS score is uncommon, with lower rates in the literature than in Keynote 048. Furthermore, Kaur et al. demonstrated significant interspecimen variability in CPS score between primary tumors and nodal metastases. This suggests the need for additional testing, for example in the case of small biopsies or negative-low results. Unfortunately, no data have been validated on which best correlates with clinical results [130].

In the last update of Keynote 048 by Harrington et al. [129], survival benefit appears independent from p16/HPV status, while patients with recurrent-only disease and no metastatic sites present a significant benefit in survival in patients treated only with pembrolizumab and chemotherapy and not with pembrolizumab monotherapy.

Patient factors influence treatment choice too. Survival benefit is influenced by age: no significant benefit in patients over 65 years has been found in EXTREME and Checkmate 141 trials, while in Keynote 048 a survival benefit in older patients was demonstrated only for the chemo-immunotherapy combination [128].

Platinum eligibility is a key limiting factor for treatment choice: renal function, cardiac function, hearing loss and neuropathy must always be checked before treatment [131], and since 5-fluorouracil is part of different regimens, DPYD testing is mandatory.

Autoimmune diseases are not a strict contraindication to immunotherapy per se, as different real-world and phase IV studies have demonstrated that treatment with ICIs is feasible with a number of autoimmune conditions [132,133,134]. Patients with viral hepatitis and HIV can be treated safely with anti-PD1/PDL-1 too but were excluded in most clinical trials [135,136,137,138]. Patients must also be kept in mind, with patients often expressing preference for chemo-free regimens or unease with the positioning of a central venous catheter for 5-Fluorouracil infusions.

There is still an unmet need for predictive markers to better select patients and thus improve immunotherapy results. Prophylactic antibiotics and gastric acid suppressants are associated with reduced survival in patients treated with anti-PD1/PD-L1 [139]; indeed, in a retrospective study of 3651 patients with oral cancer, the cohort that was administered antibiotics three months before or after immunotherapy presented shorter survival, possibly due to influence on gut and locoregional microbiomes, disrupting immunotherapy effectiveness [140]. Regarding peripheral blood biomarkers, a single-institution retrospective study identified higher LDH levels and absolute neutrophil count (ANC) as related to worse survival (OS and PFS), while higher LDH levels and lower lymphocyte percentages were related to worse ORR [141]. As ICIs fundamentally rely on lymphocyte reactivation, a lower lymphocyte count reflects resistance to checkpoint inhibitors, and indeed, several studies reported that a higher neutrophil-to-lymphocyte ratio is associated with worse survival. Higher LDH levels usually reflect a hypoxic environment and a more aggressive tumor and can indicate a radio- and chemo-refractory cancer [142]; Pan et al. posited that elevated LDH may therefore also represent resistance to immunotherapy [141].

#### 3.2.2. Radiomics Models

Radiomics-based models aim to predict post-chemoradiotherapy complications such as xerostomia, trismus, and hearing loss. Accurate prediction of both long-term and short-term oncological outcomes is crucial for personalized, risk-based treatment selection, especially in head and neck squamous cell carcinoma, where balancing disease control with acceptable toxicity is vital. Radiomics shows potential for predicting overall survival and metastasis risk in SCCHN patients. Predictive models for severe acute toxicity can guide treatment selection and identify patients who might benefit from additional supportive care. Models predicting late toxicities can customize treatment plans by adjusting dose distributions to protect specific organs at risk [143].

The genomic heterogeneity of aggressive tumors often appears as intratumoral spatial heterogeneity at anatomical and functional scales. Radiomics uses advanced quantitative analysis of medical images to non-invasively characterize this heterogeneity, offering important prognostic information about cancer. The aim is to condense this extensive data into simple prediction models to identify specific tumor phenotypes for improved treatment management [144].

Univariate analysis has shown that many features extracted from PET and CT images are significantly linked to the development of distant metastases, suggesting that the metastatic phenotype of tumors can be captured through quantitative image analysis. These features likely detect necrotic regions within tumors, indicating a higher risk of metastasis. Integrating radiomic variables with clinical data using logistic regression has positively influenced the prediction and prognostic assessment of locoregional recurrences and distant metastases, although clinical variables alone performed better for locoregional recurrences [145].

## 4. Conclusions

Locoregional disease after curative treatment, both as recurrence or second primary tumor, highly impacts patient prognosis due to patient characteristics, previous treatment and tumor features. On the other hand, 20–30% of patients with SCCHN will develop distant metastases in their life, and once metastatization occurs, survival is limited, with a median OS of only 10 months. The increased use of salvage surgery and reirradiation, depending on the stage and site of recurrence, are increasingly helpful, whereas locoregional treatments for distant metastasis are also becoming more widespread with high control rates, allowing oncologists to possibly delay chemotherapy. Choice of systemic treatment is challenging, having different alternatives to choose from, between chemotherapy, chemo-immunotherapy and immunotherapy alone, to the best of our knowledge. Improving risk stratification, prediction of treatment responses, and prognostic evaluations empower clinicians to plan treatment options, adjust treatment intensity, and customize cancer care for patients. This interest has fueled the growth of radiomics in head and neck cancer research, as bioimaging features are critical for outcome prediction. Various studies have underscored the benefits of combining radiomics analysis with conventional clinical predictors [22].

The large amount of information from radiology scanners can also be used in radiomic and machine learning models to improve clinical diagnosis [146].

As evident, the range of opportunities in head and neck cancer is finally increasing, thus giving all members of the multidisciplinary team new roles, even in the case of recurrence or metastases.

## Figures and Tables

**Figure 1 biomedicines-12-02080-f001:**
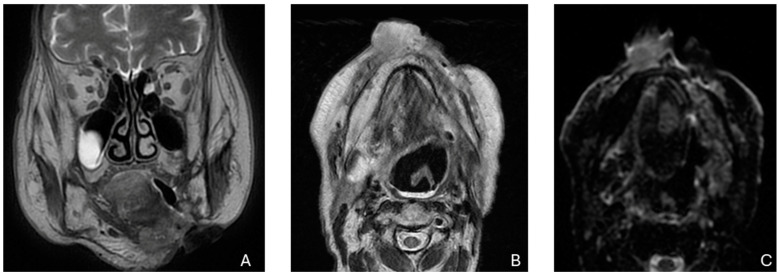
Recurrence in the anterior third of the hemitongue in a patient treated by surgery in the lower margin of the hemitongue. (**A**) The image shows a T2-weighed image, (**B**) shows a restriction signal in the diffusion-weighed image, (**C**) shows increased enhancement after intravenous injection of the contrast medium.

**Figure 2 biomedicines-12-02080-f002:**
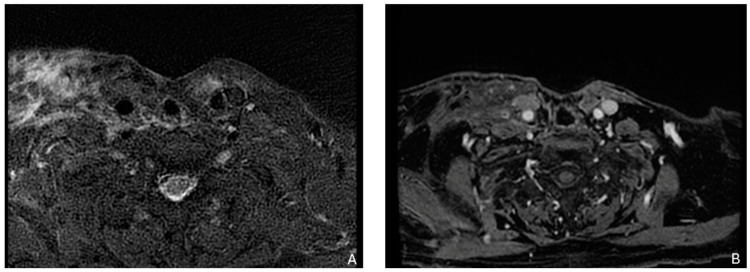
Example of a recurrence in the laterocervical right side with involvement of the thorax inlet in a patient with a larynx carcinoma treated with total laryngectomy with laterocervical dissection. (**A**) T2-weighted image. (**B**) Contrast enhancement shows that recurrence tissue is not dissociated from brachiocephalic vessels and sternocleidomastoideo muscle.

**Table 1 biomedicines-12-02080-t001:** Chemo-reirradiation.

Paper	Analysis	Treatment	Results
Langer et al., 2007 [79]	Phase II trial	RT + Cisplatin + Paclitaxel	2-year OS 25.9% 2-year PFS 15.8%
Spencer et al., 2008 [28]	Phase II trial	RT + 5-Fluorouracil + Hydroxyurea	2-year OS 15.2%, 5-year OS 3.8% Better survival in the > 1 interval from previous radiotherapy subgroup
Janot et al., 2008 [71]	Phase III trial	RT + 5-Fluorouracil + Hydroxyurea, adjuvant intent	Statistically significant benefit in DFS, no benefit in OS
Riaz et al., 2016 [73]	Retrospective analysis	RT, definite or adjuvant intent (16% + CT per physician’s choice)	2-year OS 43% 2-year LCR 47%
Tanvetyanon et al., 2009 [74]	Retrospective analysis	RT, definite or adjuvant intent	Median PFS 12.1 months median OS 19.3 months

LCR, local control rate; PFS, progression-free survival; OS, overall survival; RT, radiotherapy.

**Table 2 biomedicines-12-02080-t002:** Oligometastatic disease and local treatment.

Paper	Analysis	Treatment	Results
Bonomo et al., 2019 [100]	Retrospective analysis	SABR on lung metastasis	1-year TTP 56.2%, 2-year 35% 3moORR 75%
Bates et al., 2019 [102]	Retrospective analysis	SABR (44 out of 60 metastases were lung metastasis)	1-year OS 78%, 2-year 43% 1-year TM-LC 75%, 2-year 57%
Franzese et al., 2021 [103]	Retrospective analysis	SABR (59.1% were lung metastasis)	1-year LCR 83.1%, 2-year 70.2% 1-year OS 81%, 2-year 67.1%
Schulz et al., 2018 [106]	Retrospective analysis	SABR or surgery (63.6% were lung metastasis)	2-year OS 21.7%, 5-year 3.5%
Young et al., 2015 [101]	Metanalysis	Surgical resection of SCCHN pulmonary metastasis	5-year OS 29%
Haro et al., 2010 [105]	Retrospective analysis	Surgical resection of SCCHN pulmonary metastasis	3-year OS 53.6%, 5-year 50%
Palma et al., 2019 [111]	Phase 2 trial	SABR + SoC (systemic therapy) vs. SoC	5-year OS 42% vs. 17%, mOS 41 vs. 28 months
Sutera et al., 2019 [112]	Phase 2 trial	SABR (52.3% were lung metastasis, 10.9% were SCCHN)	5-year OS 42% in the SCCHN subgroup
Grisanti et al., 2019 [114]	Retrospective analysis	Bone radiotherapy and/or Bone directed therapy	mOS 38 months both therapies, 25 months only one, 8 months neither
Pawlik et al., 2007 [117]	Prospective study	Liver metastasis directed treated in squamous cell carcinomas (12 SCCHN)–resection, RFA or both	5-year DFS 18.6% 5-year OS 20.5%
Kurihara et al., 2021 [118]	Retrospective analysis	Hepatic resection in SCCHN or esophageal cancer	1-year OS 72%, 3-year 62% 1-year RFS 45%, 3-year 17%
Huang et al., 2014 [119]	Retrospective analysis	Partial hepatectomy vs. TACE in nasopharyngeal carcinoma	1-year OS 85.7% vs. 53.5%, 5-year 40.2% vs. 20.0% 1-year PFS 70% vs. 27%, 5-year 18% vs. 0%
Feng et al., 2022 [120]	Prospective study	Hepatic resection vs. SoC	1-year OS 86.2% vs. 61.5%, 5-year 37.3 vs. 2.9%

DFS, disease-free survival; LCR, local control rate; PFS, progression-free survival; ORR, objective response rate; OS, overall survival; SABR, stereotactic ablative radiotherapy; SCCHN, squamous cell carcinoma of head and neck; SoC, systemic therapy; TM-LC, treated metastasis local control; TTP, time to progression.

## Data Availability

No new data were created or analyzed in this study.

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
