# Peer review of "Recurrent Versus Metastatic Head and Neck Cancer: An Evolving Landscape and the Role of Immunotherapy"

_biomedicines, 2024, doi:10.3390/biomedicines12092080_

Round 1
Reviewer 1 Report
Comments and Suggestions for Authors
The review manuscript by Belfiore et. al. provides a comprehensive review of literature on recurrent versus metastatic head and neck cancer. Specifically, it focuses on the evolving landscape and the role of immunotherapy in treating recurrent versus metastatic head and neck cancer. The content of the review is extensive and covers all aspects treatments and clinical trial conducted so far. The heterogeneity and disease phenotype classification and relevant treatments are well described. The outcomes of combination of immunotherapy with radiation and chemotherapy are discussed well.
Comments on the Quality of English Language
However, the manuscript needs major editing of the language to improve the clarity and readability. Many places double negative were used in sentences (eg, lines 36 and 37). In some places, it is difficult to understand what the authors are trying to convey. Too long sentences with 4 or 5 lines and in few places one sentence forms a short paragraph (lines 153-158, 580-599). Many abbreviations are not expanded (e.g. ADC, DWI, HNSCC). SCCHN and HNSCC are interchangeably used. A table listing the immunotherapy/chemotherapy/radiation treatment options for recurrent versus metastatic head and neck cancer and highlighting the differences in treatments will be helpful to the readers.
Author Response
- However, the manuscript needs major editing of the language to improve the clarity and readability. Many places double negative were used in sentences (eg, lines 36 and 37). In some places, it is difficult to understand what the authors are trying to convey. Too long sentences with 4 or 5 lines and in few places one sentence forms a short paragraph (lines 153-158, 580-599). Text has been edited
- Many abbreviations are not expanded (e.g. ADC, DWI, HNSCC). Abbreviations have been expanded
- SCCHN and HNSCC are interchangeably used. All instances of HNSCC have been changed into SCCHN
- A table listing the immunotherapy/chemotherapy/radiation treatment options for recurrent versus metastatic head and neck cancer and highlighting the differences in treatments will be helpful to the readers. A table has been added for the main studies on chemo-reirradiation and local treatment of oligometastatic disease
Reviewer 2 Report
Comments and Suggestions for Authors
The review article entitled ‘Recurrent versus metastatic head and neck cancer: an evolving landscape and the role of immunotherapy’ was well received.
The aim of the review as mentioned by authors ‘The aim of our review the characteristics of recurrent SCCHN, based on different recurrence sites, and metastatic disease; we will also explore the possibilities not only of salvage surgery and reirradiation but also of systemic therapy choices and locoregional treatment for metastatic SCCHN’ does not coincide with the title of the article.
Although mentioned in the title, the article does not have a separate section for the role of immunotherapy nor does provide sufficient information within the text related to mentioned point.
The figures mentioned in the paper contain MRI and other data. What is the data source? Do the authors provide their own data? Or is this data from some other papers? There is no reference mentioned with the figures. The reference should be mentioned because these are not the usual figures of a review article and cannot be drawn by software like biorender or others.
In the author contributions ‘Author Contributions: For research articles with several authors, a short paragraph specifying their individual contributions must be provided. The following statements should be used “Conceptualization, M.P.B. and M.F..; methodology, M.P.; software, M.D; validation, M.F., M.P.B. and S.C.; for-mal analysis, V.N.; investigation, I.D,F.S.; resources, C.B; data curation, S.F.; writing—original draft 665preparation, M.P.; writing—review and editing, M.P.; visualization, M.F; supervision, M.P.B; All 666authors have read and agreed to the published version of the manuscript.”…… I think the authors should carefully see this. This is a simple copy-paste
Author Response
- The aim of the review as mentioned by authors ‘The aim of our review the characteristics of recurrent SCCHN, based on different recurrence sites, and metastatic disease; we will also explore the possibilities not only of salvage surgery and reirradiation but also of systemic therapy choices and locoregional treatment for metastatic SCCHN’ does not coincide with the title of the article. Although mentioned in the title, the article does not have a separate section for the role of immunotherapy nor does provide sufficient information within the text related to mentioned point. The paragraphs regarding immunotherapy in the systemic setting have been separated in a specific subsection (3.2.1. The role of immunotherapy). As immunotherapy is up to now only approved in the palliative treatment of R/M SCCHN, no separate subsection has been added to the curative intent treatment of locoregional recurrences; however, the paragraph reporting recent trials adding ICIs to chemoradiotherapy has been extended.
- The figures mentioned in the paper contain MRI and other data. What is the data source? Do the authors provide their own data? Or is this data from some other papers? There is no reference mentioned with the figures. The reference should be mentioned because these are not the usual figures of a review article and cannot be drawn by software like biorender or others. Images are provided by author professor M.P. Belfiore. A note in each comment has been added to clarify this aspect.
- In the author contributions ‘Author Contributions: For research articles with several authors, a short paragraph specifying their individual contributions must be provided. The following statements should be used “Conceptualization, M.P.B. and M.F..; methodology, M.P.; software, M.D; validation, M.F., M.P.B. and S.C.; for-mal analysis, V.N.; investigation, I.D,F.S.; resources, C.B; data curation, S.F.; writing—original draft 665preparation, M.P.; writing—review and editing, M.P.; visualization, M.F; supervision, M.P.B; All 666authors have read and agreed to the published version of the manuscript.”…… I think the authors should carefully see this. This is a simple copy-paste It was an error. Text has been edited
Reviewer 3 Report
Comments and Suggestions for Authors
An interesting overview of the different scenarios and characteristics of locoregional recurrence and metastatic disease in head and neck cancers.
Some points to focus on:
1. Nowadays, MRI represents the best diagnostic imaging for head and neck tumors. It is better to define the recurrence and its extent.
2. When describing locoregional recurrence in different subsites, the first goal is to know the previous treatment and the motivating choice. Upfront surgery or Rt +/-Cht. Why? There is a big difference in planning the second treatment. A paragraph should describe the pros and cons of the two different choices and scenarios
3. Comparative analysis of molecular patterns as markers to differentiate recurrence from new primitive is not cited. In future, these markers could help us.
4. In laryngeal recurrence, should be enfasized that the recurrence susceptible to partial laryngectomy are considerably smnaller, thus justifying the best oncological outcome. The two procedures are not comparable.
5. In laryngeal recurrence is cited the fistula rate in partial laryngectomy. NO. Pharyngocutaneous fistula it's a typical problem of total laryngectomy. After Rt tretament the typical treatment is Total laryngectomy with increasing risk of fistula score. A salvage sirgical treatmetn with partial laryngectomy is a rare event and a typical complication is a not functional organ, persisting trachetomy and pexia problem, NOT a fistula risk.
6. I think images related to disease recurrence would be more suitable. Post-conservative treatment and post-surgical treatment in different subsites.
7. it is possible to reduce the discussion on metastatic disease and chemoreirradiation with some tables reporting the different studies.
Comments on the Quality of English LanguageNot extensive challenge.
Author Response
- Nowadays, MRI represents the best diagnostic imaging for head and neck tumors. It is better to define the recurrence and its extent. Text has been edited
- When describing locoregional recurrence in different subsites, the first goal is to know the previous treatment and the motivating choice. Upfront surgery or Rt +/-Cht. Why? There is a big difference in planning the second treatment. A paragraph should describe the pros and cons of the two different choices and scenarios A paragraph has been added
- Comparative analysis of molecular patterns as markers to differentiate recurrence from new primitive is not cited. In future, these markers could help us. A paragraph has been added
- In laryngeal recurrence, should be emphasized that the recurrence susceptible to partial laryngectomy are considerably smaller, thus justifying the best oncological outcome. The two procedures are not comparable. A sentence has been added underlining this issue.
- In laryngeal recurrence is cited the fistula rate in partial laryngectomy. NO. Pharyngocutaneous fistula it's a typical problem of total laryngectomy. After Rt treatment the typical treatment is Total laryngectomy with increasing risk of fistula score. A salvage surgical treatment with partial laryngectomy is a rare event and a typical complication is a not functional organ, persisting tracheotomy and pexia problem, NOT a fistula risk. The sentence has been deleted.
- I think images related to disease recurrence would be more suitable. Post-conservative treatment and post-surgical treatment in different subsites. Figures have been modified.
- It is possible to reduce the discussion on metastatic disease and chemoreirradiation with some tables reporting the different studies. Text has been slightly edited, but as part of the focus of this paper is on systemic treatment and reirradiation and also considering Reviewer 2 comment on the absence of a separate section on immunotherapy, we did not reduce discussion on these subjects. Nevertheless, a table has been added for the main studies on oligometastatic disease.
Round 2
Reviewer 2 Report
Comments and Suggestions for Authors
Thank you for your feedback